# Evaluating the Validity of International Standards of Height, Weight, and Body Mass Index on Jordanian Children and Adolescents

**DOI:** 10.3390/healthcare12131295

**Published:** 2024-06-28

**Authors:** Walid Al-Qerem, Ruba Zumot, Anan Jarab, Judith Eberhardt, Fawaz Alasmari, Alaa Hammad

**Affiliations:** 1Department of Pharmacy, Faculty of Pharmacy, Al-Zaytoonah University of Jordan, Amman 11733, Jordan; 202127091@std-zuj.edu.jo (R.Z.); alaa.hammad@zuj.edu.jo (A.H.); 2College of Pharmacy, Al Ain University, Abu Dhabi 112612, United Arab Emirates; anan.jarab@aau.ac.ae; 3AAU Health and Biomedical Research Center, Al Ain University, Abu Dhabi 112612, United Arab Emirates; 4Department of Clinical Pharmacy, Faculty of Pharmacy, Jordan University of Science and Technology, Irbid 22110, Jordan; 5Department of Psychology, School of Social Sciences, Humanities and Law, Teesside University, Borough Road, Middlesbrough TS1 3BX, UK; j.eberhardt@tees.ac.uk; 6Department of Pharmacology and Toxicology, College of Pharmacy, King Saud University, Riyadh 12372, Saudi Arabia; ffalasmari@ksu.edu.sa

**Keywords:** BMI-for-age, growth standards, height-for-age, weight-for-age

## Abstract

Background: the variations in a child’s overall body shape and figure among different countries are attributable to differences in genetics, environmental factors, and the interaction between these elements. This study aims to evaluate the validity, reliability, and appropriateness of applying international growth standards to Jordanian children and adolescents aged 2–19 years old. Methods: 65,828 Jordanian children and adolescents (43% males; 57% females) aged 2–19 years old were selected from the Hakeem Program database and various private schools across Jordan. Height-for-age, weight-for-age, and body mass index (BMI)-for-age were analyzed comparatively for Jordanian children and adolescents against international growth standards. The z-score for each record was computed based on international equations. Results: Mean z-scores for height-for-age, weight-for-age, and BMI-for-age for both genders showed significant deviation from international standards across most age intervals. It was found that in most age groups, Jordanian children and adolescents were shorter and lighter than CDC and WHO standards, except for females at ages ≥ 16 years, who were heavier with higher BMI-for-age values than CDC standards based on weight-for-age and BMI-for-age equations. Moreover, Jordanian males at ages ≥ 12 years had lower BMI-for-age values than CDC standards. Conclusions: Jordanian children and adolescents showed significant deviations in their measurements from international standards and growth reference values. The development of a population-specific growth chart is highly recommended to enhance the accuracy of evaluating children’s and adolescents’ wellness.

## 1. Introduction

Children’s growth and development are indicative of their overall well-being. Thus, understanding children’s growth and development patterns is crucial for ensuring their healthy growth and ability to thrive [1]. Growth references are widely used and serve as valuable tools in assessing children’s health and well-being. They help mitigate disputes regarding abnormal growth and development, foster healthy growth and development, and aid in the detection of underlying diseases [2]. It is critical to highlight that variations in a child’s overall body shape and figure among different countries are attributable to differences in genetics, environmental factors, and the interaction between these elements [3].

In 2000, the Centers for Disease Control and Prevention (CDC) established growth standards based on anthropometric data regularly collected from the National Health and Nutrition Examination Survey (NHANES) between 1963 and 1994. This data comprised five cross-sectional nationally representative health surveys of US children. The 2000 CDC growth charts are the recommended standards for assessing the growth and physical development of children aged 2 through 19 years in the US [4]. CDC growth charts effectively represent the ethnic diversity and growth patterns of both breastfed and formula-fed infants in the US and these charts have been extended for global use, applicable to children and adolescents up to the age of twenty years [4].

In 2006, the WHO presided over the Multicenter Growth Reference Study Group (MGRS) with the aim of developing universal growth standards. This effort was based on data collected over six years (1997–2003) from six countries: Oman, the US, Ghana, India, Brazil, and Norway [5]. MGRS eligibility criteria include healthy growing children five years of age and younger from a variety of ethnic groups who lived in situations where there are no restrictions on thriving; breastfeeding for a minimum of the first year of age; initiation of solid feeding at the age of six months as recommended by the MGRS; full birth term; and no perinatal smoking [6]. Several limitations were encountered, which led to the NCHS not being as reliable and accurate in assessing children’s growth at early ages [7]. Therefore, in 2007, the WHO announced a new growth reference for children and adolescents aged 5 to 19 years. This was based on a reconstruction of the 1977 National Center for Health Statistics (NCHS)/WHO reference. These new references are appropriate and highly consistent with the 2006 WHO growth reference for children aged 0–5 years and the recommended cutoff values for overweight and obesity for the age group of 5–19 years [8].

Several countries worldwide have adopted and implemented the CDC growth standards, often without considering variations in children’s ethnic groups, milk feeding practices, and socioeconomic factors. In contrast, several countries tend to use population-specific growth standards to enhance precision and accuracy [9]. For example, the United Kingdom has implemented new age-based growth charts by integrating WHO growth references with their national data, resulting in a national UK–WHO growth chart tailored to their population. This chart reflects their genetic, socioeconomic, and cultural backgrounds [7,9]. Similarly, Iran, China, Saudi Arabia, and Germany have developed their own national growth standards, which provide optimal guidance and accurately reflect the growth patterns that best fit their populations [10,11,12,13]. 

Previous research conducted in Jordan found significant differences between infants and children under the age of 2 and global standards [14]. Thompson (2021) argued that frequent updates to CDC international growth chart standards are advisable, given that they are based on measurements taken more than 40 years ago and the high likelihood of these standards being outdated, and considering that children’s growth patterns and secular trends change over time [15]. Therefore, the objective of the present study was to evaluate the validity, reliability, and appropriateness of applying CDC growth charts to Jordanian children and adolescents aged 2–19 years. Should it be determined that international standards do not align with Jordanian growth patterns, then it is advisable that national growth charts and equations specifically tailored to the Jordanian population are developed. 

## 2. Materials and Methods

Data comprising anthropometric measures were retrieved from Jordanian children and adolescents across various governorates and regions from March 2023 to November 2023. The collected data encompassed both cross-sectional and longitudinal types, covering records of children and adolescents from the years 2012 to 2023. The data were obtained from the Hakeem Program, a non-profit national electronic health initiative designed to provide autonomous solutions for enhancing Jordan’s healthcare system. Additional data regarding children’s anthropometric measurements, taken in September 2022, were randomly collected from private schools from various governorates across Jordan. The measurements were taken and documented annually at the beginning of each academic year. This study followed the Declaration of Helsinki’s ethical guidelines. Ethical approval for this study was granted by the ethical committees of Al-Zaytoonah University of Jordan (REF#2023-2022/12/30) and the Jordanian Health Ministry (REF#2023/3/22).

The Ministry of Health (via Hakeem datasets) and the private schools in Jordan that participated in this study measure children’s weight and height according to international guidelines, including ISAK standards [16]. For instance, measurement techniques and physical examinations were performed by a physician and a well-trained nurse. Body weight was objectively measured using calibrated digital scales, with participants standing unassisted in lightweight clothing and without shoes. For height measurements, calibrated stadiometers were used. The children stood barefoot, upright with their heels together and level with the ground in the Frankfort plane [17]. The results were rounded to the nearest 0.1 unit, and measurements were repeated to ensure accuracy [18]. 

The obtained data consisted of anthropometric measurements, including weight and height, gender, medical history, date of birth, and date of measurement, enabling the precise calculation of the age of each participant at the time of measurement. Additionally, BMI was calculated by dividing the weight in kilograms by the height in meters squared. As per CDC selection criteria, the chosen data pertain exclusively to healthy Jordanian children and adolescents who are medically sound, not suffering from health conditions, and not undergoing any treatments that could hinder normal growth patterns, such as severe infection; chronic kidney disease (CKD); Down’s syndrome; attention deficit hyperactivity disorder; cancer; heart disease; lung disease; cystic fibrosis; Crohn’s disease; type 1 diabetes; hormonal disturbances; any stress-related conditions; hormonal therapy such as corticosteroids, Growth hormones (GHS), or gonadotropin-releasing hormone agonists; insulin; and amphetamines or any other stimulants. In addition, participants with missing or invalid data values for anthropometric measurements were excluded from the study. 

Data-cleaning procedures were applied to rectify or eliminate inaccuracies, incompleteness, corruption, malformation, duplications, and outliers or so-called biologically implausible values (BIVs) in the dataset. Outliers were identified using z-scores for height-for-age, weight-for-age, and BMI-for-age. According to CDC and WHO guidelines, defining outliers involves calculating “flags” for every anthropometric parameter. Any z-score value exceeding +5 or falling below −5 is considered an outlier and should be eliminated from the dataset [19]. Outliers are values that significantly deviate from the rest of the dataset, potentially compromising its validity and reliability. These anomalies may result from erroneous data entry, incorrect measurement techniques, and improper data coding, or they could represent extreme cases of obesity or underweight in children. Therefore, z-scores were produced for each child based on the study data and international equations, and any records that violated the acceptable z-score ranges were excluded from the analysis. 

The following flowchart demonstrates the sample retrieval and exclusion scheme (Figure 1).

### 2.1. Statistical Analysis

Statistical analysis was performed using IBM SPSS Statistics, Version 26.0 [20], and RStudio, Version 4.3.1 [21]. To ensure consistency, all measurements in children and adolescents were transformed into standard units: age was calculated in days and years, height was transformed to centimeters (cm), weight was transformed to kilograms (kg), and BMI was calculated in kg/m^2^. Additionally, data were subcategorized by gender and distributed into consecutive age intervals of two-year increments to allow for age- and gender-based analysis. Detailed age distribution that summarizes the count of participants at each age interval for each gender is available in Figure 2.

### 2.2. Validation of International Standards

Height-for-age, weight-for-age, and BMI-for-age were analyzed comparatively for Jordanian children and adolescents, aged 2–19 years, against CDC and WHO growth standards [4,22]. Using the packages ‘childsds’, ‘who anthro,’ and ‘who anthroplus’ in the R program, the z-score for each record was computed based on CDC and WHO equations for all age groups except for WHO z-scores for weight-for-age in children older than ten years, as they were not available. To assess the significance of the differences between the z-scores and the global standards, Cohen’s criteria were applied to determine the effect size and its practical importance. Cohen’s criteria classify the effect size based on the value of the z-scores. Values lower than ±0.2 are considered compatible with CDC growth standards. Specifically, values of ±0.2 represent a small effect size, indicating a slight deviation from CDC norms. Values of ±0.5 signify a medium effect size, reflecting moderate deviation. Meanwhile, values of ±0.8 or larger suggest a large deviation from CDC growth standards.

Q-Q plots were composed to evaluate the distribution of z-scores produced by the CDC and WHO for the three anthropometric measures evaluated in the study, and the plots indicated that the data were not normally distributed. Therefore, nonparametric tests were performed to compare the two standards using a Related-Samples Wilcoxon Signed Rank Test and Spearman’s rho correlations. 

## 3. Results

In total, 65,828 Jordanian children and adolescents (43% males; 57% females) aged 2 to 19 years met the eligibility criteria and were included in the study. The primary variables obtained were age (in years), weight (in kilograms), height (in centimeters), and the corresponding calculated BMI (body mass index) (in kg/m^2^). 

Figure 3 illustrates the age-related distribution between males and females in terms of mean weight (kg), mean height (cm), and mean BMI (kg/m^2^).

### 3.1. Jordanian Children’s and Adolescents’ Growth Compared with CDC Standards

Males’ height-for-age and weight-for-age values fell below CDC standards across all age intervals. Large deviations were found in age groups 12 to <14 years, 14 to <16 years, and 16 to <18 years, in which the mean height-for-age z-scores were −1.75, −2.09, and −1.75, respectively. The mean weight-for-age z-scores were −1.32, −1.53, and −1.23, respectively. BMI-for-age values for age groups 10 to <12 years and younger and 18 to ≤19 years were compatible with CDC standards. A mild deviation was found in the remaining age groups of 12 to <14 years, 14 to <16 years, and 16 to <18 years, in which the mean z-scores for BMI-for-age were −0.41, −0.49, and −0.32, respectively.

Females’ height-for-age measurements fell below CDC standards across all age groups. The highest deviation was found in age groups 12 to <14 years (mean height-for-age z-score = −1.64) and 14 to <16 years (mean height-for-age z-score = −1.36). 

Females displayed lower weight-for-age relative to CDC standards up to the age group of 14 to <16 years. The largest deviation occurred in the age group of 12 to <14 years, with significant deviations indicated by mean z-scores of −0.91. For the remaining age groups of 16 to <18 years and 18 to ≤19 years, a mild deviation was found, with values above CDC standards, with mean weight-for-age z-scores of 0.32 and 0.34, respectively.

Concerning BMI-for-age, females exhibited values that were almost comparable to CDC standards up to the older age groups of 16 to <18 years and 18 to ≤19 years. At these ages, they showed higher BMI-for-age values than the CDC standards, with mean z-scores of 0.64 at 16 to <18 years and 0.60 at 18 to ≤19 years.

### 3.2. Jordanian Children’s and Adolescents’ Growth Compared with 2006/2007 WHO Standards

The new WHO standards, which are applicable for monitoring and evaluating children’s growth and development from birth to five years, were employed to assess the growth of Jordanian children aged 2 to 5 years. The height-for-age in male children and adolescents deviated below WHO standards for all age groups. The largest deviation was seen in age groups 12 to <14 years (mean height-for-age z-score = −1.98), 14 to <16 years (mean height-for-age z-score = −2.28), and 16 to <18 years (mean height-for-age z-score = −1.8). The smallest deviations were seen in the age group 2 to <4 years (mean height-for-age z-score −0.45).

Moreover, males’ mean weight-for-age was similar to WHO standards in the youngest age group of 2 to <4 years. It mildly deviated in the age intervals of 4 to <6 years, as the sample fell below the WHO standard for weight-for-age. Moderate deviations were observed in older age groups, specifically 6 to <8 years and 8 to <10 years, with mean weight-for-age z-scores of −0.56 and −0.55, respectively.

Males’ mean BMI-for-age showed scores that were comparable to, or that only slightly deviated from, WHO standards. A deviation with higher BMI values, with a mean BMI-for-age z-score of 0.37, was seen in the youngest age group of 2 to <4 years. Lower BMI values were observed in the age groups of 12 to <14 years and 14 to <16 years, with mean BMI-for-age z-scores of −0.33 and −0.48, respectively.

For female children, the mean height-for-age was lower than WHO standards at all age intervals, indicating that Jordanian females were shorter than those in the WHO standards. The highest deviations were recorded in age groups 12 to <14 years and 14 to <16 years, with mean z-scores of −1.7 and −1.29, respectively. The smallest deviations were observed in the age group of 2 to <4 years, with a mean z-score of −0.46.

Weight-for-age was comparable to WHO standards in age groups 2 to <4 years and 8 to <10 years, with mean weight-for-age z-scores of −0.04 and −0.14, respectively. Age groups 4 to <6 years and 6 to <8 years displayed small deviations as they were thinner compared to WHO standards, with mean z-scores of −0.41 and −0.32, respectively.

For BMI-for-age, females had higher values than the WHO reference in most age groups. Moderate deviations were observed only at age intervals of 16 to <18 years and 18 to ≤19 years, with mean z-scores of 0.74 and 0.78, respectively. Mild deviations were noted at age intervals of 2 to <4 years and 8 to <10 years, with mean z-scores of 0.35 and 0.37, respectively, while the remaining age intervals were compatible with WHO standards.

Figure 4, Figure 5 and Figure 6 compare the values of mean z-scores calculated using CDC and WHO standards. 

A strong correlation was found between the CDC and WHO regarding height-for-age, weight-for-age, and BMI-for-age in both genders. For males, the values of the correlation coefficients were 0.996, 0.995, and 0.993, respectively, with a *p*-value < 0.001. Females’ correlation coefficients were 0.992, 0.993, and 0.993, respectively, with a *p*-value < 0.001. However, the Related-Samples Wilcoxon Signed Rank Test indicated that there were significant differences in all anthropometric measures in both sexes between the two international standards (with *p*-values < 0.001).

## 4. Discussion

Age- and gender-specific growth charts are essential clinical tools for monitoring a child’s appropriate longitudinal growth [14]. Due to the rapid growth rates of adolescents and children, standard weight, height, and BMI will vary annually and differ between genders. Therefore, age-appropriate percentiles and thresholds are essential for children and adolescents, and these differ significantly from the thresholds used for newborns or adults [10].

CDC growth charts are national standards that describe the general development and growth patterns of US children and adolescents from ages 2 through 20 years and are recommended to be used as international references [4]. CDC growth references use enhanced data inputs and statistical methods, providing a more accurate representation of children and adolescents across all racial and ethnic groups in the United States. These references also combine data from both formula-fed and breastfed children [4]. While CDC growth charts accurately depict the growth patterns of US children, there is ongoing debate regarding their reliability and validity when applied in Jordan to assess the growth and development of Jordanian children and adolescents [10]. 

As assumed by the WHO, growth standards achieved using the MGRS are exceptionally effective, reliable, and robust technical tools for evaluating the well-being of newborns and young children and can be used worldwide for healthy growing children from multiple ethnic, social, economic, and cultural backgrounds [5]. The 2006 WHO growth charts are designed to determine the height-for-age, weight-for-age, BMI-for-age, and weight-for-height for children up to five years old. In 2007, complementing the previous standards, the WHO introduced new growth standards for children and adolescents aged 5 to 19 years [8]. 

To the best of the authors’ knowledge, this is the first study to validate the use of CDC and WHO growth standards for Jordanian children and adolescents aged 2–19 years and to determine whether these international standards are suitable and reliable for decision-making in the context of the Jordanian population. A key element in ensuring the validity and reliability of the results depends on the precise matching of the Jordanian population with the reference populations. Our findings reveal significant variations in growth patterns between Jordanian children and CDC and WHO standards. This variation highlights the theoretical importance of context-specific growth standards that account for genetic, environmental, and socioeconomic differences across populations. These results challenge the prevailing assumption that international growth charts are universally applicable, suggesting instead that localized assessments may be more accurate and beneficial.

From a practical standpoint, the demonstrated discrepancies highlight the need for developing national growth charts for Jordan. Such tailored tools will empower healthcare professionals to make more accurate health assessments and interventions tailored to the specific growth trends observed within the local pediatric population. This approach is essential for the early detection and management of growth-related health issues, ensuring that interventions are based on accurate, relevant data.

The findings of the present study showed strong agreement between CDC and WHO standards as both displayed a clear deviation from CDC standards in terms of gender-related height-for-age and weight-for-age. Specifically, the height-for-age values indicated that Jordanian children and adolescents consistently had lower measurements, evidenced by uniformly low mean z-scores across all age intervals for both genders. As a result, the height of Jordanian children and adolescents was found to be below the benchmarks set by both standards, indicating that Jordanian children and adolescents are shorter than the reference population. In terms of weight-for-age values, Jordanian males showed consistently lower readings, with all mean z-scores indicating measurements below the normative standards set by CDC and WHO standards. On the other hand, Jordanian females exhibited weight-for-age values below the international CDC standards until the age of 16 years. Starting from the age of 16 years and older, the mean z-score values for females showed a slight positive deviation from CDC standards, indicating that they were heavier than the universal standards. BMI-for-age values were comparable to CDC standards at young ages (until 10 to <12 years in males and 14 to <16 years in females). Beyond these ages, BMI-for-age measurements in males were below CDC standards while they were above in females. The findings of the current study align with those of a previous national study conducted in 2001, which investigated the growth status of Jordanian school children. That study also observed distinct variations in children’s weight-for-age and height-for-age patterns, noting that Jordanian children fell below CDC standards. One of the suggested reasons for the lower than CDC reference norms observed in the study may be attributed to the prevalence of consanguineous marriages, which are common in Jordanian culture. The authors advocated for Jordanian health authorities to establish local standards in order to correctly evaluate Jordanian children’s growth and development [23].

The study results align with the findings of previous studies that examined the validity of using CDC and/or WHO standards across different populations. For example, a cross-sectional descriptive study conducted in Calabar, Nigeria, in 2020 tested the validity of using CDC standards on the local population. The results indicated that the anthropometric parameters (height, weight, and BMI) measured for Nigerian children did not consistently align with CDC growth standards at most ages, leading to recommendations for national growth charts for Nigeria [24].

Another study published in 2019 aimed to objectively compare the BMI percentile curves of Iranian children and adolescents, aged 6 to 18 years, from various regions in Iran with WHO and CDC standards. It found statistically significant differences, indicating that Iranian children and adolescents generally have lower BMI values compared to the reference populations. These results prompted the development of population-specific BMI growth charts with new cutoff points based on the collected data, to better categorize the weight status of Iranian children and adolescents [10]. Additionally, in Delhi, a cross-sectional study compared the impact of applying WHO growth standards to the interpretation of growth parameters in school children aged 8–15 years. By calculating the height-for-age and BMI-for-age z-scores for each participant, significant differences were seen, and the application of nationally derived growth standards was highly recommended [25]. Additionally, several systematic reviews showed that none of the studies reviewed reported a growth trend comparable to WHO standards across all considered growth indicators. This emphasizes the importance of designing nation-specific growth standards [7,26].

It is important to keep in mind that growth variation exists within every population, making it difficult to develop a universal growth standard that accurately represents the growth rates of all nations [7]. Several factors may account for the differences in growth patterns between Jordanian and American children and adolescents over time, including genetic characteristics, nutritional intake, and eating habits, as well as environmental and socioeconomic conditions. Other factors that affect growth patterns are the general health status of the population and differences in the prevalence of growth-related illnesses among groups, such as infections or malnutrition. Differences in growth patterns may also stem from variations in urbanization and lifestyle. For instance, malnutrition and limited access to healthcare services such as prenatal care and immunization programs tend to be more prevalent in rural areas. Malnutrition has a negative impact on the mental, physical, and behavioral growth and development of many children [27].

As indicated by the high correlation coefficient, there was a high level of agreement in the trend of the z-scores produced by both the CDC and WHO for all the anthropometric measures. Nevertheless, there were significant differences in the z-scores produced by the two equations. This was also reported in previous studies that found that significant differences are present in growth status indicators and z-score values for both sexes in different age groups, which may be attributed to differences in study design and the characteristics of the enrolled samples [7,26].

The insights from the present study pave the way for further research into the application of growth standards in other unique demographic settings, potentially leading to a broader reassessment of the suitability of current international growth charts. Moreover, these findings could inform policy revisions and support legislative efforts to develop and implement localized growth monitoring tools, ultimately enhancing child health surveillance and intervention strategies.

Several limitations of the present study need to be acknowledged. Firstly, no standard measurement techniques were applied; although qualified nurses took the readings and entered them into the Hakeem database, neither the accuracy of the data entry nor the consistency of the measurements taken by several nurses was evaluated. Secondly, there was no representation of geographical and socioeconomic data, which limited the interpretation of their effect on the patterns of growth and development. Lastly, the results are limited by the nature of the study; the cross-sectional part prevented any attempt to investigate causality between the variables, while the longitudinal part of the study did not include the whole sample and was not measured at the same age intervals.

However, the current study has several key strengths. A major strength that significantly advances our understanding of growth and development patterns in Jordanian children and adolescents is its comprehensive and nationally representative design. Data were collected both cross-sectionally and longitudinally, encompassing all regions of the kingdom, from south to north. The sample included participants from various social and economic levels. In addition, the large sample size allowed for stable estimates and precise measurements at the outer percentiles. This helped minimize selection bias and significantly enhanced the reliability of the results. An additional strength of the study lies in the comprehensive details available in the database regarding the children’s medical conditions. This enabled the exclusion of participants who were considered unhealthy or on medications potentially affecting their growth. Consequently, a more accurate and true reflection of the growth patterns of Jordanian children and adolescents can be confidently presented. 

## 5. Conclusions

Significant differences were observed between the growth measurements of Jordanian children and adolescents and international growth standards in this study. Jordanian children and adolescents were generally shorter and lighter than international standards, except for females above the age of 16, who exhibited higher BMI values, and males above the age of 12, who had lower BMI values. This suggests a potential limitation in applying a single universal growth norm to children and adolescents. Developing and adopting Jordanian-specific growth charts would greatly enhance the monitoring of growth and development for Jordanian children and adolescents, significantly improving healthcare services and outcomes.

## Figures and Tables

**Figure 1 healthcare-12-01295-f001:**
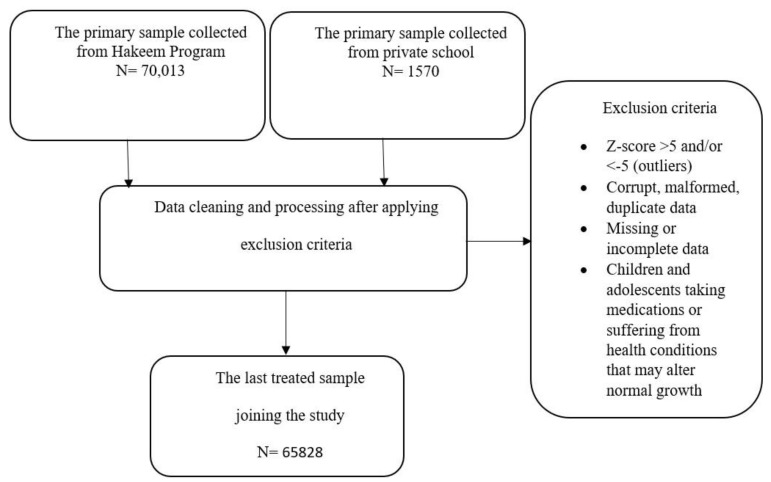
Flowchart illustrating the sample enrollment.

**Figure 2 healthcare-12-01295-f002:**
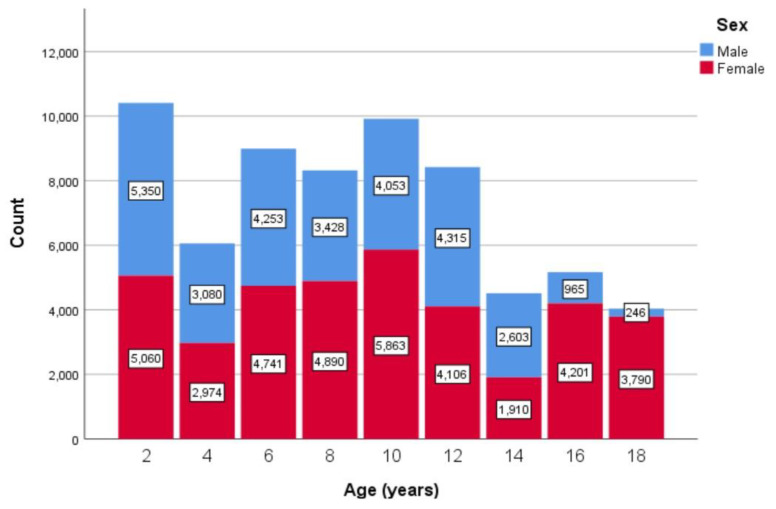
Jordanian participants at each age interval (years).

**Figure 3 healthcare-12-01295-f003:**
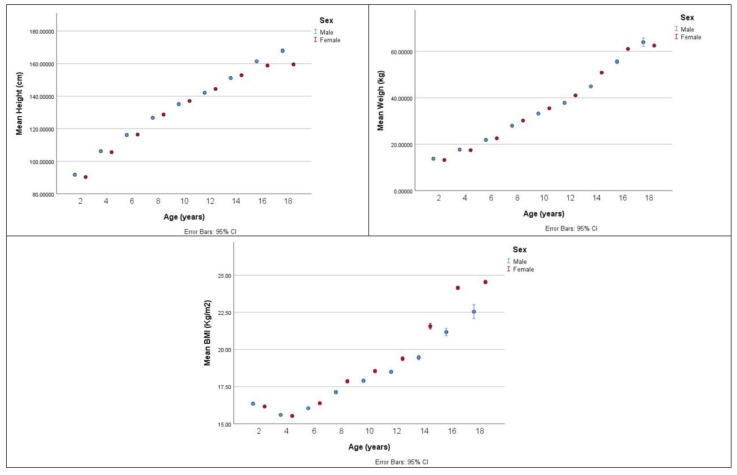
Mean height (cm), mean weight (kg), and mean BMI (kg/m^2^) by age and by sex.

**Figure 4 healthcare-12-01295-f004:**
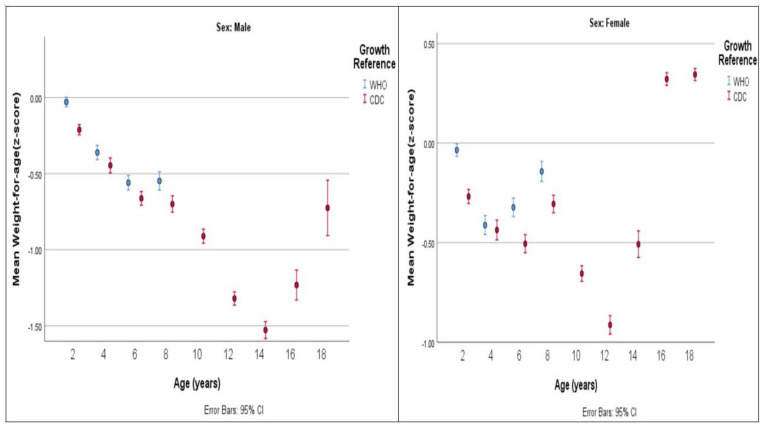
Mean weight-for-age z-score by age group (years) and by growth reference.

**Figure 5 healthcare-12-01295-f005:**
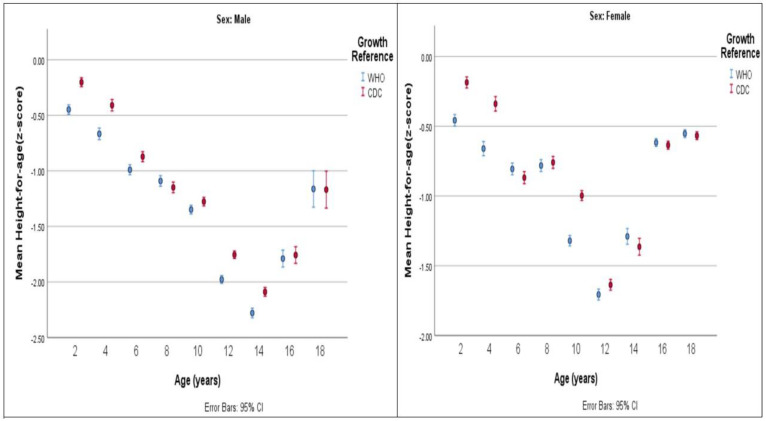
Mean height-for-age z-score by age group (years) and by growth reference.

**Figure 6 healthcare-12-01295-f006:**
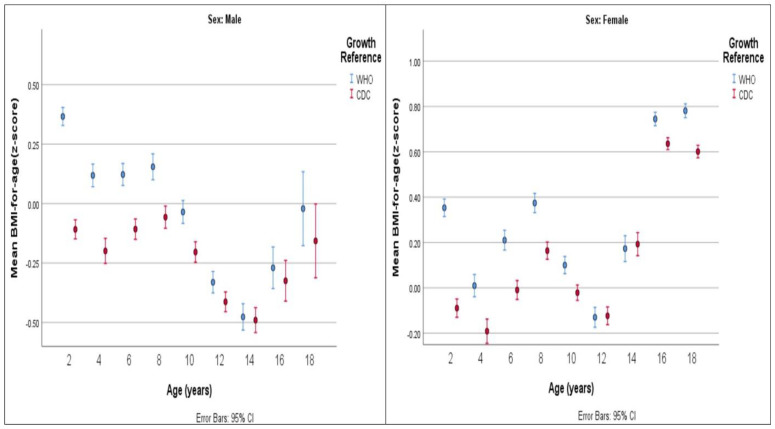
Mean BMI-for-age z-score by age group (years) and by growth reference.

## Data Availability

The datasets analyzed during the current study are available at https://doi.org/10.5281/zenodo.12520372.

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
