# Peer review of "Evaluating the Validity of International Standards of Height, Weight, and Body Mass Index on Jordanian Children and Adolescents"

_healthcare, 2024, doi:10.3390/healthcare12131295_

Round 1
Reviewer 1 Report
Comments and Suggestions for Authors
The article works with a large sample and uses two standards to verify anthropometric measurements of children/young people of both sexes and ages between 2-5 years. The article came with comments, which may suggest areas for improvement. As critical contributors, authors must take excellent care to ensure that the submitted version is free of specific problems, furthering the advancement of our collective knowledge.
As an evaluator, I have some comments that need to be noted by the authors:
1- Statistical analysis is elementary. The authors need to explore the data in depth. There is no comparison (via statistical testing) of the values ​​from the two standards used (WHO and CDC). It is essential to highlight that any comparison via statistical tests requires meeting assumptions for using the test.
2—The authors abuse the use of large tables. Graphs would be much more recommended, indicating values ​​and confidence intervals on the X-axis and age on the Y-axis. The graph would visually facilitate the evolution of values ​​depending on the age of the children/young people.
3—The study's potential to influence standards, legislation, or products needs a more explicit discussion of its practical and theoretical implications. From a theoretical perspective, what novel insights does this study offer? What gap does it fill in the existing body of research? These discussions would provide a more comprehensive understanding of the study's potential impact and future research directions, inspiring the authors to continue their work.
4—The references are mostly old. Reveals that the authors did not conduct a severe bibliographical survey to locate the primary updated references. It also explains the authors' difficulty in clarifying the justification and relevance of this study from a scientific and practical point of view.
Author Response
Comment 1: The article works with a large sample and uses two standards to verify anthropometric measurements of children/young people of both sexes and ages between 2-5 years. The article came with comments, which may suggest areas for improvement. As critical contributors, authors must take excellent care to ensure that the submitted version is free of specific problems, furthering the advancement of our collective knowledge.
As an evaluator, I have some comments that need to be noted by the authors:
Response 1: Thank you for your time, efforts and comments that significantly improved the quality of the manuscript
Comment 2: Statistical analysis is elementary. The authors need to explore the data in depth. There is no comparison (via statistical testing) of the values ​​from the two standards used (WHO and CDC). It is essential to highlight that any comparison via statistical tests requires meeting assumptions for using the test.
Response 2: Thank you for your comment. A comparison between the two international standards was performed as you suggested. The following was added to the method section: “Q-Q plots were composed to evaluate the distribution of z-scores produced by the CDC and WHO for the three anthropometric measures evaluated in the study, and the plots indicated that the data was not normally distributed. Therefore, nonparametric tests were performed to compare the two standards using Related-Samples Wilcoxon Signed Rank Test, and Spearman’s rho correlations.“
The following was added to the results section: “A strong correlation was found between the CDC and WHO regarding height-for-age, weight-for-age, and BMI-for-age in both genders. For males, the values of the correlation coefficients were 0.996, 0.995, and 0.993 respectively with a p-value < 0.001. Females’ correlation coefficients were 0.992, 0.993, and 0.993 respectively with a p-value < 0.001. However, the Related-Samples Wilcoxon Signed Rank Test indicated that there were significant differences in all anthropometric measures in both sexes between the two international standards (with p-values < 0.001).”
The following was added to the discussion section: “As indicated by the high correlation coefficient, there was a high level of agreement in the trend of the z-scores produced by both the CDC and WHO for all the anthropometric measures. Nevertheless, there were significant differences in the z-scores produced by the two equations. This was also reported in previous studies that found that significant differences are present in growth status indicators and z-score values for both sexes in different age groups, which may be attributed to differences in study design and the characteristics of the enrolled samples [7, 26].”
Comment 3: The authors abuse the use of large tables. Graphs would be much more recommended, indicating values ​​and confidence intervals on the X-axis and age on the Y-axis. The graph would visually facilitate the evolution of values ​​depending on the age of the children/young people.
Response 3: The tables were converted into figures as suggested.
Comment 4: The study's potential to influence standards, legislation, or products needs a more explicit discussion of its practical and theoretical implications. From a theoretical perspective, what novel insights does this study offer? What gap does it fill in the existing body of research? These discussions would provide a more comprehensive understanding of the study's potential impact and future research directions, inspiring the authors to continue their work.
Response 4: Thanks for this suggestion. The following paragraphs have been added to the Discussion:
“Our findings reveal significant variations in growth patterns between Jordanian children and the CDC and WHO standards. This variation highlights the theoretical importance of context-specific growth standards that account for genetic, environmental, and socioeconomic differences across populations. These results challenge the prevailing assumption that international growth charts are universally applicable, suggesting instead that localized assessments may be more accurate and beneficial.”
“From a practical standpoint, the demonstrated discrepancies highlight the need for developing national growth charts for Jordan. Such tailored tools will empower healthcare professionals to make more accurate health assessments and interventions tailored to the specific growth trends observed within the local pediatric population. This approach is essential for the early detection and management of growth-related health issues, ensuring that interventions are based on accurate, relevant data.”
“The insights from the present study pave the way for further research into the application of growth standards in other unique demographic settings, potentially leading to a broader reassessment of the suitability of current international growth charts. Moreover, these findings could inform policy revisions and support legislative efforts to develop and implement localized growth monitoring tools, ultimately enhancing child health surveillance and intervention strategies.”
Comment 5: The references are mostly old. Reveals that the authors did not conduct a severe bibliographical survey to locate the primary updated references. It also explains the authors' difficulty in clarifying the justification and relevance of this study from a scientific and practical point of view.
Response 5: The references were updated where possible, as suggested.
Reviewer 2 Report
Comments and Suggestions for Authors
In this article the validity, reliability, and appropriateness of applying the CDC international growth standards to Jordanian children and adolescents aged 2-20 years are evaluated. The significant deviation from the international standards across most age intervals suggests the importance of formulating national population-specific growth charts.
I have several minor remarks which need addressing prior to publication.
Line 91 not clear: Data cleaning was applied to conducted to rectify or eliminate inaccuracies
Figure 1: why “inclusion criteria” are mentioned here?
Line 128 not clear: Values lower than ±0.2 are deemed
Line 133: the comma should become colon, and the following sentence is too long.
Line 182. The comma should become dot.
Lines 199-201: the sentence should be rephrased.
Lines 207-216: the sentence is too long.
Line 221: add a comma after “31 months”.
Line 223: add a comma after “months”.
Line 231: the comma after “standards” should become colon.
Line 233: the colon should be eliminated.
Line 236: add a comma after “Concerning weight-for-age”.
Line 249: years should become “months”.
I have no further suggestions to improve the paper. It is a very well written comprehensive analysis and will make an important contribution to the field.
Thank you for the opportunity to review this paper.
Author Response
Comment 1: In this article the validity, reliability, and appropriateness of applying the CDC international growth standards to Jordanian children and adolescents aged 2-20 years are evaluated. The significant deviation from the international standards across most age intervals suggests the importance of formulating national population-specific growth charts.
I have several minor remarks which need addressing prior to publication.
Response 1: Thank you for your time and efforts
Comment 2: Line 91 not clear: Data cleaning was applied to conducted to rectify or eliminate inaccuracies
Response 2: Thanks for pointing this out. This has now been revised to read: “Data cleaning procedures were applied to rectify or eliminate inaccuracies”.
Comment 3: Line 128 not clear: Values lower than ±0.2 are deemed
Response 3: This has been revised as follows: “Values lower than ±0.2 are considered compatible with CDC growth standards.”
Comment 4: Line 133: the comma should become colon
Response 4: Corrections were made.
Comment 5: the following sentence is too long.
Response 5: The sentence has been revised as follows: “Additional validation tests were conducted for children aged 2 to 5 years using the ‘who anthro’ package. This age group was divided into one- to two-month age intervals for detailed analysis. The analysis involved comparing the mean and 95 percent confidence limits (CL) for z-scores for height-for-age, weight-for-age, weight-for-height, and BMI-for-age against the WHO growth standards.”
Comment 6: Line 182. The comma should become dot.
Response 6: Corrections were made.
Comment 7: Lines 199-201: the sentence should be rephrased.
Response 7: This has been rephrased as follows: “The new WHO standards, which are applicable for monitoring and evaluating children's growth and development from birth to five years, were employed to assess the growth of Jordanian children aged 2 to 5 years.”
Comment 8: Lines 207-216: the sentence is too long.
Line 221: add a comma after “31 months”.
Line 223: add a comma after “months”.
Line 231: the comma after “standards” should become colon.
Line 233: the colon should be eliminated. Corrections were made.
Line 236: add a comma after “Concerning weight-for-age”. Corrections were made.
Line 249: years should become “months”.
Response 8: Corrections were made.
Comment 8: I have no further suggestions to improve the paper. It is a very well written comprehensive analysis and will make an important contribution to the field.
Thank you for the opportunity to review this paper.
Response 9: Thank you for your support and encouragement
Reviewer 3 Report
Comments and Suggestions for Authors
First of all, the reviewer would like to thank the authors for their work and efforts in trying to improve heatlth science healthknowledge.
The objective of this reading is, to evaluate the validity, reliability, and appropriateness of applying the CDC growth charts to Jordanian children and adolescents between the ages of 2-20 years.
I consider it an interesting paper but it needs some modifications to be taken into consideration for publication.
General:
- Authors should remove comments that they left forgotten.
Abstract:
- Authors should not put (1)(2).. to separate each section.
- To indicate that they found differences in almost all age ranges, could the authors make the effort to comment on the direction of any of these variables? For example, Jordanian Children present a lower weight than the standards?
Methods:
- Why between 2 and 20: children under 18 have percentile scores that are not homologous to those over 18. In fact, the WHO establishes percentiles by age up to 18 years.
- What international references are they analyzing? What are the dates of the references and measurements they collected? Also, why did they take the 2000 CDC and not the 2006 WHO or the 1977 NCHs? There are these three standards, and they should be discussed in your paper or indicated in the methods section as the results do discuss the WHO 2-5 years, when the standards also include up to 18 years of age.
- It would also be necessary for them to further define the methods of evaluation of the measurements that were taken in the schools. For example, was the height taken according to ISAK standards, where the height is taken in traction in the Frankfur plane... Or something like that. Or the procedures equal to those of the international reference data.
- It would be necessary to define the databases from which the results were extracted and the schools from which they are analyzed. How were the data taken, what procedures were followed, who measured them... Were the measurement methods the same as those followed in the international references with which they were compared?
Author Response
Comment 1: First of all, the reviewer would like to thank the authors for their work and efforts in trying to improve heatlth science healthknowledge.
The objective of this reading is, to evaluate the validity, reliability, and appropriateness of applying the CDC growth charts to Jordanian children and adolescents between the ages of 2-20 years.
I consider it an interesting paper but it needs some modifications to be taken into consideration for publication.
Response 1: Thank you for your time, efforts and comments that significantly improved the quality of the manuscript
General:
Comment 2: Authors should remove comments that they left forgotten.
Response 2: Corrections were made.
Abstract:
- Comment 3: Authors should not put (1)(2).. to separate each section.
Response 3: Corrections were made.
Comment 4: To indicate that they found differences in almost all age ranges, could the authors make the effort to comment on the direction of any of these variables? For example, Jordanian Children present a lower weight than the standards?
Response 4: Thank you for your comment. The following modification was made to the abstract section for further clarification: “Results: Mean z-scores for height-for-age, weight-for-age, and BMI-for-age for both genders showed significant deviation from the international standards across most age intervals. It was found that in most age groups, Jordanian children and adolescents were shorter and lighter than the CDC and WHO standards, except for females at ages ≥ 16 years which were heavier with higher BMI-for-age values than CDC standards based on weight-for-age and BMI-for-age equations. Moreover, Jordanian males at ages ≥ 12 years had lower BMI-for-age values than the CDC standards.”
Methods:
Comment 5: Why between 2 and 20: children under 18 have percentile scores that are not homologous to those over 18. In fact, the WHO establishes percentiles by age up to 18 years.
Response 5: Thank you for your comment. The WHO growth standards cover the ages from 2-19 years, while the CDC covers the ages between 2-20 years. Therefore, the authors deleted the data for participants older than 19 and retained the data for all the younger age groups.
Comment 6: What international references are they analyzing? What are the dates of the references and measurements they collected?
Response 6: The dates of the references and measurement collected are written in the materials and methods part: “Data comprising anthropometric measures were retrieved from Jordanian children and adolescents across various governorates and regions from March 2023 to November 2023. The collected data encompassed both cross-sectional and longitudinal types, covering records of children and adolescents from the years 2012 to 2023”.
Comment 7: Also, why did they take the 2000 CDC and not the 2006 WHO or the 1977 NCHs? There are these three standards, and they should be discussed in your paper or indicated in the methods section as the results do discuss the WHO 2-5 years, when the standards also include up to 18 years of age.
Response 7: Comparison against WHO standards for children 5-19 years was added to the study. For clarification, the following paragraph was added in the introduction section: “In 2006, the WHO presided over the Multicenter Growth Reference Study Group (MGRS) with the aim of developing universal growth standards. This effort was based on data collected over six years (1997-2003) from six countries — Oman, the US, Ghana, India, Brazil, and Norway [4]. MGRS eligibility criteria include healthy growing children five years of age and younger from a variety of ethnic groups who lived in situations where there are no restrictions on thriving; breastfeeding for a minimum of the first year of age, initiation of solid feeding at the age of six months as recommended by the MGRS, full birth term, and no perinatal smoking [5]. Several limitations were encountered, which led to the NCHS not being as reliable and accurate in assessing children's growth at early ages [6]. Therefore, in 2007, the WHO announced a new growth reference for children and adolescents aged five to 19 years. This was based on the reconstruction of the 1977 National Center for Health Statistics (NCHS)/WHO reference. These new references are appropriate and highly consistent with the 2006 WHO growth reference for children aged 0-5 years and the recommended cutoff values for overweight and obesity for the age group of 5-19 years [7].”
Additionally, the following was added to the Methods section: “Height-for-age, weight-for-age, and BMI-for-age were analyzed comparatively for Jordanian children and adolescents, aged 2-19 years, against the CDC and WHO growth standards [8, 9]. Using the packages ‘childsds’, ‘who anthro’ and ‘who anthroplus’ in the R program, the z-score for each record was computed based on CDC and WHO equations for all age groups except for the WHO z-scores for weight-for-age in children older than ten years, as they are not available”.
Comment 8: It would also be necessary for them to further define the methods of evaluation of the measurements that were taken in the schools. For example, was the height taken according to ISAK standards, where the height is taken in traction in the Frankfur plane... Or something like that. Or the procedures equal to those of the international reference data.
Response 8: Thank you for your comment. The following has been added: “The Ministry of Health (via Hakeem datasets) and the private schools in Jordan that participated in this study measure children's weight and height according to international guidelines, including ISAK standards [10]. For instance, the measurement techniques and physical examinations were performed by a physician and a well-trained nurse. Body weight was objectively measured using calibrated digital scales, with participants standing unassisted in lightweight clothing and without shoes. For height measurements, calibrated stadiometers were used. The children stood barefoot, upright with their heels together and level with the ground in the Frankfort plane [11]. The results were rounded to the nearest 0.1 unit, and measurements were repeated to ensure accuracy [12].
Comment 9: It would be necessary to define the databases from which the results were extracted and the schools from which they are analyzed. How were the data taken, what procedures were followed, who measured them... Were the measurement methods the same as those followed in the international references with which they were compared?
Response 9: The following has been included in the Materials and Methods section:
- “Data comprising anthropometric measures were retrieved from Jordanian children and adolescents across various governorates and regions from March 2023 to November 2023. The collected data encompassed both cross-sectional and longitudinal types, covering records of children and adolescents from the years 2012 to 2023. The data were obtained from the Hakeem Program, a non-profit national electronic health initiative designed to provide autonomous solutions for enhancing Jordan’s healthcare system. Additional data regarding children's anthropometric measurements, taken in September 2022, were randomly collected from private schools from various governates across Jordan. The measurements were taken and documented annually at the beginning of each academic year.”
- “The Ministry of Health (via Hakeem datasets) and the private schools in Jordan that participated in this study measure children's weight and height according to international guidelines, including ISAK standards [10]. For instance, the measurement techniques and physical examinations were performed by a physician and a well-trained nurse. Body weight was objectively measured using calibrated digital scales, with participants standing unassisted in lightweight clothing and without shoes. For height measurements, calibrated stadiometers were used. The children stood barefoot, upright with their heels together and level with the ground in the Frankfort plane[11]. The results were rounded to the nearest 0.1 unit, and measurements were repeated to ensure accuracy [12]."
Reviewer 4 Report
Comments and Suggestions for Authors
The authors of the paper “Evaluating the validity of international standards of Height, weight, and body mass index on Jordanian children and adolescents” studied the applicability of the CDC and WHO growth charts to Jordanian children and adolescents between the ages of 2-20 years.
I recommend including the age-related distribution of anthropometric measurements (such as height, weight, and BMI) and the prevalence of body mass index categories based on gender and age in the article. This addition would provide valuable insights into the demographic characteristics of the study population and enhance the comprehensiveness of the findings.
In the Discussion section, it would be beneficial to compare the results with other similar studies evaluating the accuracy of growth charts for children. For instance, there are systematic reviews discussing this topic (Oliveira MH, Pereira DDS, Melo DS, Silva JC, Conde WL. Accuracy of international growth charts to assess nutritional status in children and adolescents: a systematic review. Rev Paul Pediatr. 2022 Apr 4;40:e2021016. doi: 10.1590/1984-0462/2022/40/2021016. PMID: 35442268; PMCID: PMC8983011).
In line 54, please explain the meaning of “methods of milk feeding”. Did you refer to “milk feeding practices”?.
In line 57, it is stated that “In contrast, several countries tend to use population-specific 55 growth standards to enhance precision and accuracy”. It would be beneficial to give some exemples.
I recommend considering transforming the data presented in Table 1 into a graph or chart format to enhance visual clarity and facilitate easier interpretation for readers.
I recommend revising the English language to enhance clarity and readability. For instance, in line 22: instead of “children and adolescents between the ages of 2-20 21 years”, please use “children and adolescents aged 2 to 21 years old”.
The conclusions drawn in the article appear to be overly general. It would be beneficial for the authors to provide more specific and nuanced conclusions based on the findings presented in the study.
Author Response
Comment 1: The authors of the paper “Evaluating the validity of international standards of Height, weight, and body mass index on Jordanian children and adolescents” studied the applicability of the CDC and WHO growth charts to Jordanian children and adolescents between the ages of 2-20 years.
Response 1: Thank you for your time, efforts and comments that significantly improved the quality of the manuscript
Comment 2: I recommend including the age-related distribution of anthropometric measurements (such as height, weight, and BMI) and the prevalence of body mass index categories based on gender and age in the article. This addition would provide valuable insights into the demographic characteristics of the study population and enhance the comprehensiveness of the findings.
Response 2: Thank you for the suggestion, this information was added in Figure 3.
Comment 3: In the Discussion section, it would be beneficial to compare the results with other similar studies evaluating the accuracy of growth charts for children. For instance, there are systematic reviews discussing this topic (Oliveira MH, Pereira DDS, Melo DS, Silva JC, Conde WL. Accuracy of international growth charts to assess nutritional status in children and adolescents: a systematic review. Rev Paul Pediatr. 2022 Apr 4;40:e2021016. doi: 10.1590/1984-0462/2022/40/2021016. PMID: 35442268; PMCID: PMC8983011).
Response 3: Thank you for your comment. The reference was added and the following was added to the Discussion section: “The study results align with the findings of previous studies that examined the validity of using CDC and/or WHO standards across different populations. For example, a cross-sectional descriptive study conducted in Calabar, Nigeria, in 2020 tested the validity of using CDC standards on the local population. The results indicated that the anthropometric parameters (height, weight, and BMI) measured for Nigerian children did not consistently align with the CDC growth standards at most ages, leading to recommendations for national growth charts for Nigeria [13].
Another study published in 2019 aimed to objectively compare the BMI percentile curves of Iranian children and adolescents, aged 6 to 18 years, from various regions in Iran with the WHO and CDC standards. It found statistically significant differences, indicating that Iranian children and adolescents generally have lower BMI values compared to the reference populations. These results prompted the development of population-specific BMI growth charts with new cutoff points based on the collected data, to better categorize the weight status of Iranian children and adolescents [14]. Additionally, in Delhi, a cross-sectional study compared the impact of applying WHO growth standards on the interpretation of growth parameters in school children aged 8-15 years. By calculating the height-for-age and BMI-for-age z-scores for each participant, significant differences were seen, and the application of nationally derived growth standards was highly recommended [15]. Additionally, several systematic reviews showed that none of the studies reviewed reported a growth trend comparable to WHO standards across all considered growth indicators. This underscores the importance of designing nation-specific growth standards [3, 6].”
Comment 4: In line 54, please explain the meaning of “methods of milk feeding”. Did you refer to “milk feeding practices”?.
Response 4: The sentence was changed as suggested.
Comment 5: In line 57, it is stated that “In contrast, several countries tend to use population-specific 55 growth standards to enhance precision and accuracy”. It would be beneficial to give some examples.
Response 5: Thank you for your comment. The following statements were added: “For example, the United Kingdom has implemented new age-based growth charts by integrating WHO growth references with their national data, resulting in a national UK-WHO growth chart tailored to their population. This chart reflects their genetic, socioeconomic, and cultural backgrounds [6, 16]. Similarly, Iran, China, Saudi Arabia, and Germany have developed their own national growth standards, which provide optimal guidance and accurately reflect the growth patterns that best fit their populations [14, 17-19].
Comment 6: I recommend considering transforming the data presented in Table 1 into a graph or chart format to enhance visual clarity and facilitate easier interpretation for readers.
Response 6: The table was replaced with a bar chart, as suggested.
Comment 7: I recommend revising the English language to enhance clarity and readability. For instance, in line 22: instead of “children and adolescents between the ages of 2-20 21 years”, please use “children and adolescents aged 2 to 21 years old”.
Response 7: This has been modified as suggested, and additional corrections have been made throughout the manuscript, to enhance clarity and readability.
Comment 8: The conclusions drawn in the article appear to be overly general. It would be beneficial for the authors to provide more specific and nuanced conclusions based on the findings presented in the study.
Response 8: The conclusion section has been modified as follows:
“Significant differences were observed between the growth measurements of Jordanian children and adolescents and the international growth standards in this study. Jordanian children and adolescents were generally shorter and lighter than the international standards, except for females above the age of 16, who exhibited higher BMI values, and males above the age of 12, who had lower BMI values. This suggests a potential limitation in applying a single universal growth norm to children and adolescents. Developing and adopting Jordanian-specific growth charts would greatly enhance the monitoring of growth and development for Jordanian children and adolescents, significantly improving healthcare services and outcomes.”
Round 2
Reviewer 1 Report
Comments and Suggestions for Authors
The authors invested a lot of effort into improving the article. All recommendations were met, and several corrections were made. The article was significantly enhanced, presenting less speculative results, excellent quality, and scientific rigor.
The article did not present severe problems that would prevent its publication. I appreciate the opportunity to contribute to improving this article.
Reviewer 3 Report
Comments and Suggestions for Authors
I congratulate and thank the authors for the work done with the suggested changes. I consider that the work is now ready for publication.
Reviewer 4 Report
Comments and Suggestions for Authors
All my comments have been addressed in the revised manuscript and I recommend its publication.